# Understanding the Role of Equivariance in Self-supervised Learning

**Yifei Wang**[*]
MIT

**Kaiwen Hu**[*]
Peking University

**Sharut Gupta**
MIT

**Ziyu Ye**
The University of Chicago

**Yisen Wang**
Peking University

**Stefanie Jegelka**
TUM[†] and MIT[‡]

## Abstract

Contrastive learning has been a leading paradigm for self-supervised learning, but it is widely observed that it comes at the price of sacrificing useful features (*e.g.,* colors) by being invariant to data augmentations. Given this limitation, there has been a surge of interest in equivariant self-supervised learning (E-SSL) that learns features to be augmentation-aware. However, even for the simplest rotation prediction method, there is a lack of rigorous understanding of why, when, and how E-SSL learns useful features for downstream tasks. To bridge this gap between practice and theory, we establish an information-theoretic perspective to understand the generalization ability of E-SSL. In particular, we identify a critical explaining-away effect in E-SSL that creates a synergy between the equivariant and classification tasks. This synergy effect encourages models to extract class-relevant features to improve its equivariant prediction, which, in turn, benefits downstream tasks requiring semantic features. Based on this perspective, we theoretically analyze the influence of data transformations and reveal several principles for practical designs of E-SSL. Our theory not only aligns well with existing E-SSL methods but also sheds light on new directions by exploring the benefits of model equivariance. We believe that a theoretically grounded understanding on the role of equivariance would inspire more principled and advanced designs in this field. Code is available at https://github.com/kaotty/Understanding-ESSL.

## 1 Introduction

Self-supervised learning (SSL) of data representations has made remarkable progress. Existing SSL methods can be categorized into two types: invariant SSL (I-SSL) and equivariant SSL (E-SSL). The idea of I-SSL is to encourage the representation to be invariant to input augmentations (*e.g.,* color jittering). Contrastive learning that pulls positive samples closer and pushes negative samples apart is widely believed to be a prominent I-SSL paradigm, leading to rapid progress in recent years [13, 50, 5, 80, 46, 30, 3, 32, 33, 60, 49, 14, 37, 15, 77, 64, 81]. Nevertheless, since invariant representations lose augmentation-related information (*e.g.,* color information), their performance on downstream tasks can be hindered, as frequently observed in practice [47, 17, 34].

---

[*]Equal Contribution. Correspondence: Yifei Wang <yifei_w@mit.edu>.

[†]CIT, MCML, MDSI

[‡]EECS, CSAIL

38th Conference on Neural Information Processing Systems (NeurIPS 2024).

In view of these limitations of I-SSL, there has been a growing interest in revisiting E-SSL. Contrary to I-SSL, E-SSL learns representations that are sensitive to (or aware of) the applied transformation.[4] For instance, RotNet [31] is an early exemplar of E-SSL that learns discriminative features by predicting the rotation angles from randomly rotated images [43]. It has also been exploited in recent works and achieves promising improvements in conjunction with I-SSL [73, 67, 17, 18, 25, 54, 34]. Recently, E-SSL has shown potential for serving as the foundation for building visual world models [26].

Despite this intriguing progress in practice, compared to invariant SSL methods with a vast literature of theoretical analyses [58, 66, 48, 35, 68, 59, 78], there is little theoretical understanding of equivariant SSL methods. A particular difficulty lies in the understanding of the pretraining tasks, which may seem quite irrelevant to downstream classification. Taking RotNet as an example, the random rotation angle is *independent* of the image class, so it is unclear how rotation-equivariant representations are helpful for image classification. Generally speaking, it is unclear **why, when, and how equivariant representations can generalize to downstream tasks**.

In view of this situation, the primary goal of this paper is not to design a new E-SSL variant, but to revisit the basic E-SSL methods and *understand* their essential working mechanisms. We fulfil this goal by proposing a simple yet theoretically grounded explanation for understanding general E-SSL from an information-theoretic perspective. We show that the effectiveness of E-SSL can be understood via the "explaining-away" effect in statistics, which implies an intriguing *synergy effect* between the image class $C$ and the equivariant transformation $A$ (*e.g.,* rotation) such that almost surely, they have strictly positive mutual information *when given the input $X$, i.e., $I(C; A|X) > 0$* that explains the effectiveness of E-SSL. This understanding also provides valuable guidelines for practical E-SSL design with three principles to pursue a large synergy effect $I(C; A|X)$: lossy transformations, class relevance, and shortcut pruning, as been validated on practical datasets. Theoretically, we also quantitatively analyze the influence of data transformation on the synergy effect with a theory model.

Equipped with these theoretical findings, we further revisit advanced E-SSL methods in the recent literature [73, 67, 17, 18, 25, 54, 34] and find that many of these empirical successes can be well explained in our framework from two aspects: finer equivariance and multivariate equivariance. Besides, motivated by our theory, we also discover an under-explored aspect of E-SSL, model equivariance, where we show that adopting equivariant neural networks can yield strong improvements for certain E-SSL methods. These fruitful theoretical and practical merits suggest that our E-SSL theory provides a general and practically useful explanation for understanding and designing E-SSL methods that have the potential to guide future E-SSL designs.

## 2 Related Work

**Invariant and Equivariant SSL.** Without access to labels, SSL methods design various surrogate tasks that create self-supervision for representation learning. Early SSL methods, often in the form of predictive learning, learn from predicting the transformation of randomly transformed images, such as, RotNet [31], Jigsaw [51], Relative Patch Location [20]. Later, discriminating instances in the latent space with contrastive learning demonstrates prominent performance [21, 72, 52, 40, 13, 38, 56], with variants including non-contrastive methods [33], clustering methods [10–12], regularization methods [77]. However, data augmentations used in contrastive learning to avoid shortcuts often come at the cost of information lost for downstream tasks (*e.g.,* color for flower classification). To address this issue, there is a surge of interest in E-SSL that learns features to be sensitive to the applied transformations. Among them, Xiao et al. [73] use separate embeddings for each augmentation. Wang et al. [67] apply equivariant prediction on residual vectors between positive views. Dangovski et al. [17] combine contrastive learning and rotation prediction. Devillers and Lefort [18], Garrido et al. [24] utilize conditional predictors with augmentation parameters. Park et al. [54], Gupta et al. [34] model latent equivariant transformations explicitly.

**Theory of SSL.** Most existing theories of SSL methods focus on contrastive learning (CL) and its variants from different perspectives: information maximization [52, 40, 65], downstream generalization [58, 66, 48, 35, 68, 59], feature dynamics [66, 69], asymmetric designs [63, 82], feature identifiability [79], *etc.* But for general E-SSL methods, there is little, if any, theoretical understanding

---

[4]A rigorous definition of feature equivariance requires more restrictions; for example, the transformations must be invertible. In existing SSL literature, equivariance is (loosely) referred to as the property that features are aware of general input transformations (not necessarily invertible). We follow this convention in this paper.

on how they learn meaningful features for downstream tasks (in particular, image classification). Our work fills this gap by establishing a general information-theoretic framework for understanding E-SSL.

**Equivariance in Deep Learning.** Invariance and equivariance represent data symmetries that can be exploited during learning. There are two approaches to utilize invariance and equivariance. One is *equivariant learning* (to which E-SSL belongs) that uses equivariant training regularization such that features are *approximately* equivariant; the other is equivariant models that obey *exact* equivariance by design *w.r.t.* groups like rotation and scaling [16, 29, 8]. Equivariant models find wide applications in graph, manifold, and molecular domains [8, 29], but are rarely explored for equivariant SSL. In this work, we find that model equivariance can be particularly helpful for equivariant learning in terms of both training and generalization, which opens an interesting direction to explore on the interplay between equivariant learning and equivariant models for future research.

## 3 The Challenges of Understanding Equivariant SSL

**Notations.** We introduce existing SSL methods from a probabilistic perspective. Generally, we denote a random variable by a capital letter such as $X$, its sample space as $\mathcal{X}$, and its outcome as $x$. We learn a representation $(Z/\mathcal{Z}/z_x)$ from the input $(X/\mathcal{X}/x)$ through a deterministic encoder function $F\colon \mathcal{X} \to \mathcal{Z}$. The general goal of SSL is to learn discriminative representations that are predictive of the image classes (labels) without actual access to label information. For ease of discussion, we mainly adopt the common Shannon information, where the entropy of $X$ is $H(X) = -\mathbb{E}_{P(X)} \log P(X)$ and the mutual information between $X$ and $Y$ is $I(X;Y) = H(X) - H(X|Y)$. It is also tempting to adopt $\mathcal{V}$-information [75] that is analogous to Shannon's notion but aligns better with practice by taking into account computational constraints. For ease of understanding, we adopt Shannon's information in the main paper and extend the main results to $\mathcal{V}$-information in Appendix C.

**Equivariant SSL (E-SSL).** For each raw input $\bar{X}$ sampled from the training set $\mathcal{D}$, we independently draw a random augmentation $A$ and get the augmented sample $X = T(\bar{X}, A)$ with a transformation mapping $T\colon \bar{\mathcal{X}} \times \mathcal{A} \to \mathcal{X}$. The general objective of Equivariant SSL (E-SSL) is to learn representations $Z = F(X)$ that are sensitive to the applied transformation $A$. For example, RotNet [31] utilizes random four-fold rotation $\mathcal{A} = \{0°, 90°, 180°, 270°\}$ for data augmentation, and learns feature equivariance by predicting the rotation angles from the representation $Z$. Therefore, E-SSL is driven by maximizing the following mutual information between the augmentation $A$ and the representation $Z$:

$$\max\ I(A;Z). \tag{1}$$

Note again that equivariance studied in this paper, as in many E-SSL works [17, 18, 24, 34], is a relaxed notion of exact equivariance defined in a group-theoretical sense (with invertibility and compositionality) as in equivariant networks [16]. In E-SSL, equivariance generally means augmentation sensitivity, and the mutual information $I(A;Z)$ measures the degree of equivariance of $Z$ to $A$.

**Contrastive Learning as E-SSL.** Contrary to E-SSL, I-SSL enforces features to be invariant to the applied augmentation $A$. CL is widely believed to be an example of invariant learning [17]. In CL, we apply two random data augmentations, $A_1, A_2$ to the same input $\bar{X}$ and get two positive samples $X_1, X_2$ and their representations $Z_1, Z_2$ respectively. Since CL is driven by pulling $Z_1, Z_2$ together, their mutual information objective is often formalized as $\max_{Z_1=F(\bar{X},A_1),Z_2=F(\bar{X},A_2)} I(Z_1;Z_2)$ [1]. However, it is easy to observe that the constant outputs $Z = const$ are also optimal with maximal $I(Z_1;Z_2)$, suggesting that invariance alone is sufficient for SSL. In fact, contrastive learning can mitigate feature collapse with the help of pushing away from the representation of the other instances (*i.e.,* negative samples), making it essentially **an equivariant learning task *w.r.t.* the instance, known as instance discrimination** [21, 72]. Indeed, contrastive objectives are essentially non-parametric formulations of instance classification [72], and under similar designs, parametric instance classification achieves similar performance [9]. Non-contrastive variants with only positive samples are also shown to have inherent connection to contrastive methods in recent studies [63, 82, 23].

**Equivariance is *Not* All You Need.** A common intuition among E-SSL methods is that better downstream performance comes from better feature equivariance [18, 25, 54, 34]. Here, we begin our discussion by showing a counterexample in the following proposition. All proofs in this paper can be found in Appendix A.

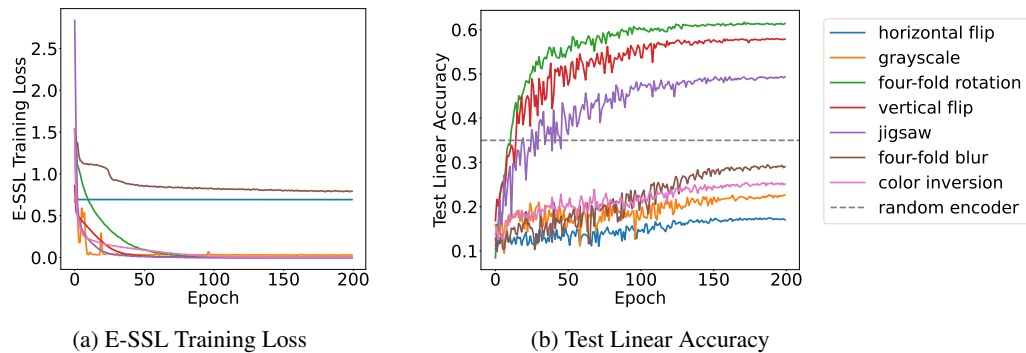

(a) E-SSL Training Loss

(b) Test Linear Accuracy

Figure 1: Comparison between different transformations for E-SSL on CIFAR-10 with ResNet-18. Note that different pretraining tasks may have different classes (*e.g.,* 4 for rotation and 2 for horizontal flip). The baseline is a random initialized encoder with 34% test accuracy under linear probing.

**Proposition 1** (Useless equivariance). *Assume that the original input $\bar{X} \in \mathbb{R}^d$ and the augmentation $A \in \mathbb{R}^{d'}$ are independent, and $X = [\bar{X}, A] \in \mathbb{R}^{d+d'}$ is obtained with direct concatenation (DC). Then, there exists a simple linear encoder that has perfect equivariance to $A$, but yields random guessing on downstream classification.*

Proposition 1 shows an extreme case when perfect equivariance is unhelpful for feature learning at all. Inspired by this finding, we further examine common image transformations for E-SSL: horizontal flip, grayscale, four-fold rotation, vertical flip, jigsaw, four-fold blur and color inversion (details in Appendix B). Prior to ours, Dangovski et al. [17] show that when merged with contrastive learning, additional equivariance to some augmentations (*e.g.,* four-fold rotation and vertical flip) can bring benefits while some are even harmful (*e.g.,* horizontal flip). It is not fully explored how much *E-SSL alone* depends on the chosen augmentation. Below, we study seven common transformations for E-SSL on CIFAR-10 [45] with ResNet-18: horizontal flip, four-fold rotation, grayscale, vertical flip, jigsaw, four-fold blur, and color inversion. We only apply random crops to avoid cross-interactions between different transformations. So the numbers reflect the relative strengths of different methods, instead of their optimal performance that can be attained.

Figure 1 reveals big differences between different choices of transformations: with linear probing, four-fold rotation and vertical flip perform the best and attain more than 60% accuracy, while the others do not even attain significant gains over random initialization (34%). This distinction cannot be simply understood via **feature usefulness**, since color information imposed by learning grayscale and color inversion is known to be important for classification [74]. Meanwhile, in Figure 1a, we find that the **degree of equivariance** (measured by the training loss of E-SSL) does not explain the difference either, since among ineffective ones, some with large training loss have very low equivariance (*e.g.,* horizontal flip), while some have very high equivariance with nearly zero equivariant loss (*e.g.,* grayscale). These phenomena show that equivariance alone, either strong or weak, does not have a good or bad indication of downstream performance, which motivates us to provide a more general understanding of E-SSL.

## 4 A Theory of Equivariant SSL

In Section 3, we have shown that feature equivariance alone does not guarantee effective downstream performance, which makes it even unclear how equivariant learning extracts useful features. To resolve these puzzles, we provide an information-theoretic analysis for E-SSL that serves as a natural explanation for the phenomena above.

### 4.1 Explaining E-SSL via Explaining-away

We start by establishing a causal diagram of the data generation process of E-SSL, where we assume that the original input $\bar{X}$ is generated from its class variable $C$ (relevant to input semantics, *e.g.,* shape), intrinsic equivariance variable $\bar{A}$ (relevant to semantics, *e.g.,* the intrinsic orientation of an object) and style variable $S$ (features irrelevant to semantics and targeted equivariance, *e.g.,* color

and texture) through some unknown processes. Then, we apply an independently drawn augmentation variable $A$ (*e.g.,* a random rotation angle), and get the transformed input $X$. Afterwards, we obtain its representation $Z$ through a neural network.

**Collider structure in E-SSL.** The causal diagram shows that the class variable $C$ and the augmentation variable $A$ are independent. However, there exists a so-called *collider* structure where the augmented sample $X$ is a common child of $C$ and $A$. A well-known fact from statistics called the *explaining-away* effect (*a.k.a.* selection bias) [55, 44] says that in a collider block, when conditioning on the collider $X$ or its descendent like $Z$, the parents $C$ and $A$ are no longer independent. For example, the weather $(A)$ and the road condition $(C)$ are independent factors that can contribute to car accidents $(X)$.

However, given that an accident happens ($X$ is known), if we know that it rains today, it would be less likely that the road is broken, and vice versa. In this case, we say that the weather $A$ explains away the possibility of road conditions $C$. The theorem below formally characterises the explain-effect effect in the E-SSL process and its information-theoretic implication. A caveat is that Lemma 1 guarantees that explaining-away happens in most, but not all cases (*e.g.,* Proposition 1), and we explain these exceptions in Section 4.2.

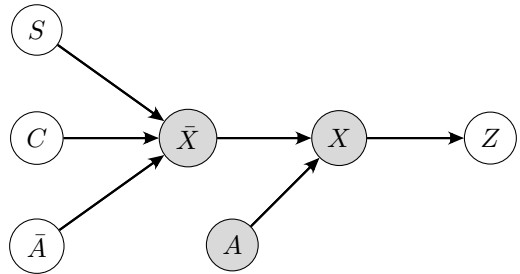

Figure 2: The causal diagram of equivariant self-supervised learning. The observed variables are in grey. $C$: class; $S$: style; $\bar{A}$: intrinsic equivariance variable; $\bar{X}$: raw input; $A$: augmentation; $X$: augmented input; $Z$: representation.

**Lemma 1** (Explaining-away in E-SSL). *If the data generation process obeys the diagram in Figure 2, then almost surely, $A$ and $C$ are not independent given $X$ or $Z$, i.e., $A \not\perp\!\!\!\perp C|X$ and $A \not\perp\!\!\!\perp C|Z$. It implies that $I(A;C|X) > 0$ and $I(A;C|Z) > 0$ hold almost surely.*

**Explaining-away helps E-SSL.** In statistics, explaining-away often appears as the selection bias in observational data that misleads causal inference (*e.g.,* the Berkson's paradox [6]) and demands careful treatment [76, 7]. In contrast, explaining-away plays a critical *positive* role in E-SSL. In particular, the fact $I(A;C|Z) > 0$ implies an important *synergy* effect between $A$ and $C$ during equivariant learning, as shown below:

$$I(A;C|Z) = H(A|Z) - H(A|Z,C) > 0 \quad \implies \quad H(A|Z) > H(A|Z,C). \tag{2}$$

Eq. (2) implies that for the same feature $Z$, using class information $C$ gives a better prediction of $A$ (lower uncertainty $H(A|Z,C)$) than without using class features. Intuitively, given a rotated image, recognizing the object class $C$ in the first place makes it easier to determine the rotation angle $A$. Driven by this synergy effect, the encoder will learn to encode class information $C$ in the representation to assist the equivariant prediction of $A$. [5] We formally characterize this intuition in the following theorem.

**Theorem 1** (Class features improve equivariant prediction). *Under the data generation process in Figure 2, consider an E-SSL task with input $X$, its class $C_X$, and its representation $Z$. Assume a class representation $Z_C = \phi(C_X)$ that can perfectly predict the label $C_X$ ($\phi$ is an invertible mapping). Then, almost surely, the combined feature $\tilde{Z} = [Z, Z_C]$ obtained by appending $Z_C$ to $Z$ will strictly improve the equivariant prediction with larger mutual information $I(A;\tilde{Z}) > I(A;Z)$. Also, we have $I(C;\tilde{Z}) \geq I(C;Z)$, so the classification performance improves in the meantime.*

As an implication of Theorem 1, to achieve better equivariant prediction, during E-SSL, the model will try to extract more class features, which will jointly improve downstream classification. This explains why during E-SSL with rotation prediction, the classification accuracy also rises along the process, outperforming the random encoder (Figure 1b).

---

[5]When taking a closer look, there could be multiple explaining-away effects in Figure 2. For example, we have $\bar{A} \to X \leftarrow A$. In practice, it could mean that knowing the intrinsic rotation of an image helps predict the rotated angle $A$. Also, we have $\bar{A} \to X \leftarrow C$, meaning that knowing $\bar{A}$ is also helpful for predicting the class $C$. This is a more nuanced analysis of how predicting rotated angles $A$ could finally help predict the class $C$.

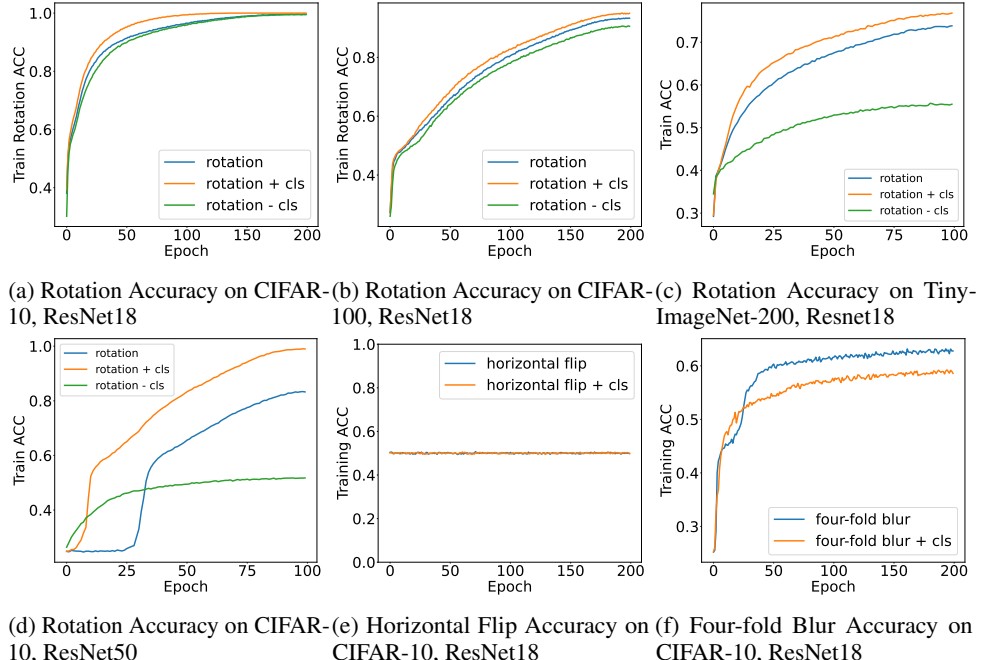

(a) Rotation Accuracy on CIFAR-10, ResNet18

(b) Rotation Accuracy on CIFAR-100, ResNet18

(c) Rotation Accuracy on Tiny-ImageNet-200, Resnet18

(d) Rotation Accuracy on CIFAR-10, ResNet50

(e) Horizontal Flip Accuracy on CIFAR-10, ResNet18

(f) Four-fold Blur Accuracy on CIFAR-10, ResNet18

Figure 3: A controlled experiment on the influence of class information on equivariant prediction. We include three methods: 1) equivariant prediction (baseline); 2) jointly minimizing equivariant and classification losses ("+cls"); 3) minimizing the equivariant loss while adversarially maximizing the classification loss [22] ("- cls"). We study rotation prediction for (a), (b), (c) and (d), horizontal flip for (e), and four-fold blur for (f).

**Remark: Extension to $\mathcal{V}$-information**. In the discussion above, we mainly adopt Shannon's entropy measures for simplicity, which ignores computational and modeling constraints. Computation-aware notions like $\mathcal{V}$-information [75] would align our theory better with practice. Notably, if we replace Shannon information with $\mathcal{V}$-information, the analyses above still hold (see Appendix C). A nice property of $\mathcal{V}$-information is that feature extraction steps can *increase* the information by making prediction computationally easier (which only decreases information in Shannon's notion instead), providing better justification for the benefit of representation learning. Thus, from the perspective of $\mathcal{V}$-information, in ESSL, class features *increase the equivariance w.r.t. $A$* with easier computation.

**Verification of synergy effects via controlled experiments.** To validate the above analysis in practice, we further carry out a *controlled experiment* to study how class information affects the equivariant pretraining task. Specifically, taking the rotation prediction task as an example, we add or substitute a class prediction loss with an additional linear head in the pretraining objective. In the former case, we explicitly inject class information into the presentation by joint training with the classification loss; in the latter, we explicitly eliminate class information from the representation by adversarially maximizing the classification loss [22] (see Appendix B). As shown in Figure 3, we get slightly better rotation prediction accuracy when explicitly incorporating the class information, while getting worse performance (with a larger margin) when discouraging class information, which agrees well with Theorem 1. Note that there is still nontrivial training accuracy because the class is not the only factor that can explain equivariant prediction (style features $S$ can also play a role).

## 4.2 Maximizing the Synergy Effect: Principles for Practical Designs of E-SSL

Our theoretical understanding above not only establishes theoretical explanations for downstream performance, but also provides principled guidelines for E-SSL design. The overall principle is to maximize the synergy $I(A; C|X) = H(A|X) - H(A|X, C)$, which can be understood from the following aspects that explain various E-SSL behaviors that we observe in Section 3.

**Principle I: "Lossy" Transformations.** First, let us look at $H(A|X)$, which determines the upper bound of the explaining-away effect. A higher $H(A|X)$ means that the equivariant prediction task

is inherently harder. Revisiting Proposition 1, our theory gives a natural and rigorous explanation for why direct concatenation (DC) fails for E-SSL. Essentially, the DC output $X = [A, \bar{X}]$ admits a simple linear encoder such that $A$ can be perfectly recovered from $X$, implying $H(A|X) = 0$, which leads to $I(A; C|X) = 0$, *i.e.,* no explaining-away effect. This implies an intriguing property of E-SSL, that in order to attain nontrivial performance on downstream tasks, *the chosen transformation $T$ must be lossy* — in the sense that one cannot perfectly infer $A$ after the transformation, *i.e.,* $H(A|X) > 0$.[6] Considering computational and model constraints in practical scenarios, this task should be at least hard for the chosen training configuration (*i.e.,* $H_{\mathcal{V}}(C|X) > 0$). Only when the transformation is hard enough, neural networks will strive to learn class information to assist its prediction. Indeed, Figure 1 shows that the transformations whose training loss decreases very quickly (*e.g.,* grayscale and jigsaw) indeed have relatively poor test accuracy, which further verifies our theory.

**Principle II: Class Relevance.** Aside from task hardness, we also need to ensure $H(A|X, C)$ is low enough; *i.e., extracting class information can effectively improve equivariant prediction.* As a negative example, with a direct concatenation $X = [C, A]$ as in Proposition 1, even if we add noise to $A$ such that $H(A|X) > 0$, extracting the class $C$ is still unhelpful for predicting $A$. From an information-theoretic perspective, it satisfies $H(A|X) = H(A|X, C)$, so we always have $I(A; C|X) = 0$. In Figure 1, horizontal flip and four-fold blur have large training losses until the end of the training, *even if we deliberately inject class features* (see Figures 3e & 3f). This suggests that these equivariant tasks are intrinsically hard and class information does not contribute much to equivariant prediction. Instead, rotation prediction and vertical flip are hard at the beginning, but the uncertainty can be decreased significantly via learning class information. These transformations thus have a large synergy effect that benefits downstream performance. We conjecture that it is because these transformations are global (*i.e.,* changing pixel positions) instead of local changes (*i.e.,* only modifying pixel values) like grayscale and color inversion, so class information as global image semantics are more helpful for such tasks. Another important implication is that the transformation should be class-preserving so as to make class features helpful for the equivariant task. This rule has been verified extensively in contrastive learning [58, 62, 35, 68].

**Principle III: Shortcut Pruning.** Note that in the causal diagram (Figure 2), class $C$ and style $S$ features jointly determine the raw input $\bar{X}$. According to our theory, style features may also explain the equivariant target $A$ (*i.e.,* $I(A; S|X) > 0$). Since style features are often easier for NN learning [28], they can become shortcuts for equivariant prediction such that class features are suppressed [28, 57]. Therefore, to ensure the learning of class-related

Table 1: Comparison of rotation prediction under different augmentations (CIFAR-10, ResNet18).

| Augmentation | Train Rot ACC | Test Cls ACC |
|---|---|---|
| None | 99.98 | 56.92 |
| Crop+flip | 97.71 | 57.32 |
| SimCLR [13] | **83.26** | **59.06** |

semantic features, it is important to avoid these shortcuts. One effective approach to corrupt these style features (to some extent) through aggressive data augmentation, *e.g.,* color jitter, cropping, and blurring commonly adopted in contrastive learning, without corrupting class features a lot. Indeed, Chen et al. [13] show that the choice of data augmentations plays a vital rule in the success of contrastive learning, and Tian et al. [62] point out its goal is to prune class-irrelevant features. Here, we generalize this principle to E-SSL as well through our explaining-away framework. As shown in Table 1, the aggressive data augmentations from SimCLR also bring much better performance for E-SSL methods, bringing RotNet (82.26%) close to SimCLR (89.49%). It demonstrates that instead of merging with contrastive learning as in all recent E-SSL works [67, 17, 18, 25, 54, 34], learning from equivariance *alone* can potentially achieve competitive performance.

### 4.3 Analysis on the Influence of Transformation

The theory in Section 4.1 guarantees that E-SSL will learn class features almost surely under general conditions. Yet, without further knowledge, it is generally hard to derive more quantitative results for downstream performance. For a concrete discussion, we consider a simplified data generation process as an exemplar. Note that simplified data models are frequently adopted in the literature of self-supervised learning theory [63, 71] to gain insights for their real-world behaviors.

---

[6]For rotation prediction, there exist samples whose rotated angle cannot be uniquely determined, such as, frogs and airplane. Thus, rotation is also a lossy transformation in this sense.

**Setup.** We consider a simple combination of the class $C$ and the augmentation $A$ as a weighted sum,

$$X = A + \lambda C, \tag{3}$$

where $\lambda \in \mathbb{R}$ is the mixing coefficient. Here, we assume a balanced class setting, where $C \sim \mathrm{Cat}(N_C)$ follows a uniform categorical variable over $N_C$ classes: $0, 1, \ldots, N_C - 1$. Similarly, we assume that the augmentation $A \sim \mathrm{Cat}(N_A)$ is an *independent* uniform categorical variable over $N_A$ classes: $0, 1, \ldots, N_A - 1$. In this simple setting, it is easy to see that given $X$, when $C$ is known, we will have a perfect knowledge of $A$ as $A = X - \lambda C$, indicating $H(A|X, C) = 0$. Therefore, we have $I(A; C|X) = H(A|X)$. In other words, transformations only influence the explaining-away effect through the uncertainty of predicting $A$. This is an extreme case for the ease of theoretical analysis. Nevertheless, the following theorem shows that under this setup, we can have a quantitative characterization of the optimal choice of $N_A$ and $\lambda$ that sheds light on the design of E-SSL methods.

**Theorem 2.** *The following results hold for the additive problem in Eq. (3):*

1) ***Balanced Mixing is Optimal.*** *With $N_C$ and $N_A$ held constant, $I(A; C|X)$ is maximized under balanced mixing with $\lambda = 1$.*

2) ***Large Action Space is Beneficial.*** *With $N_C$ held constant and $\lambda = 1$, we have a lower bound of the mutual information $I(A; C|X) \geq \ln N_C - \frac{1}{N_A} \left[ (N_C - 1) \ln N_C - \frac{(N_C - 1)^2}{N_C} \ln (N_C - 1) + \frac{N_C - 2}{2} \right]$, which is **monotonically increasing** with respect to $N_A$.*

Theorem 2 has two important implications. First, it suggests that a balanced mixing of $A$ and $C$ gives the optimal synergy effect, since it can maximize the uncertainty of using $X$ for predicting $A$ alone (agreeing with Principle I). Second, it shows that a large action space ($|\mathcal{A}|$) is preferred, making it harder to use spurious features (e.g., the boundary values of $C$ and $A$) as a shortcut to determine $A$ (agreeing with Principles II and III). These theoretical results illustrate our analyses above and provide insights for understanding advanced designs in E-SSL methods, as elaborated below.

## 5 Understanding Advanced E-SSL Designs

In Section 4, we have established a theoretical understanding of basic E-SSL through the explaining-away effect. However, basic E-SSL (like rotation prediction) often fails to achieve satisfactory performance, and many advanced designs have been proposed to enhance E-SSL performance [67, 17, 18, 25, 54, 34]. In this section, we further explain how these advanced designs improve performance by enhancing the synergy effect between class information and equivariant prediction.

### 5.1 Fine-grained Equivariance

A conclusion from Theorem 2 is that a larger action space of the transformation $A$ benefits the explaining-away effect by increasing the task difficulty $H(A|X)$. Guided by this principle, one way to improve E-SSL is through learning from more fine-grained equivariance variables with a larger action space ($|\mathcal{A}|$), which encourages models to learn diverse features and avoid feature collapse for specific augmentations. For example, four-fold rotation is a 4-way classification task while CIFAR-100 has 100 classes. When the neural networks are expressive enough such that it clusters samples with the same augmentation to (almost) the same representation (known as neural collapse [53]), the class features also degrade or vanish, which hinders downstream classification. For example, Table 1 shows that for rotation prediction, stronger augmentations suffer from less feature collapse (lower training accuracy), while enjoying better classification accuracy. Indeed, we show that the advantages of state-of-art SSL methods can be understood through this information-theoretic perspective.

**Information-theoretic Understanding of Instance Discrimination.** As disclosed in Section 3, contrastive learning is essentially an E-SSL task with equivariance prediction of instances. Specifically, each raw example $\bar{x}_i$ serves as an instance-wise class, forming an action space $\mathcal{I}$, where all augmented samples of $\bar{x}_i$ belong to the class $i$. Therefore, the instance classification task has an action space of $|\mathcal{I}| = N$, where $N$ is the number of training dataset that is much larger than rotation prediction with $|\mathcal{A}| = 4$, making instance discrimination a harder task, especially under strong data augmentations [72, 21]. Since the instance index $I$ is also *independent of the class* variable $C$, it is not

Table 2: Training rotation prediction accuracy and test linear classification accuracy under different base augmentations (CIFAR-10, ResNet18).

| Dataset | Augmentation | Network | Train Rotation ACC | Test Classification ACC | Gain |
|---------|-------------|---------|--------------------|-----------------------|------|
| CIFAR-10 | None | ResNet18 | 99.98 | 56.92 | |
| | | EqResNet18 | **100.00** | **72.32** | **+16.40** |
| | Crop&Flip | ResNet18 | 97.71 | 57.32 | |
| | | EqResNet18 | **99.97** | **82.54** | **+25.22** |
| | SimCLR | ResNet18 | 83.26 | 59.06 | |
| | | EqResNet18 | **91.98** | **82.26** | **+23.20** |

fully clear why it is helpful for learning class-relevant features.[7] Instead, our explaining-away theory gives a natural explanation from the instance classification perspective. In this way, our explanation of E-SSL can be regarded as a unified understanding of existing SSL variants.

**Equivariance Beyond Instance.** Although contrastive learning already adopts a very large action space with $|\mathcal{I}| = N$, there is recent evidence showing that it can still learn shortcuts [57, 73] and lack feature diversity [70]. Therefore, it is natural to consider even finer-grained equivariance, such as learning to predict patch-level or pixel-level features [2], inputs [39], or tokenized patches [4], which comprises many variants of SSL methods, ranging from MAE [39], BERT [19], to diffusion models [41, 61]. Here, either random mask [19] or Gaussian noise [41] can be viewed as random variables (similar to the rotated angle in rotation prediction) and is independent of the class semantics, so they fit into our theory as well. Features learned from these tasks do show more diversity in practice and benefit downstream tasks requiring fine-grained semantics [39, 42]. Therefore, our theory provides a principled way to understanding the benefits of fine-grained supervision in SSL.

## 5.2 Multivariate Equivariance

As discussed in Section 4.2, equivariant prediction may have class-irrelevant features as shortcuts, while corrupting these features (*e.g.,* color) with data augmentation might affect certain downstream tasks (*e.g.,* flower classification that requires color information too). A more principled way that has been explored recently is through joint prediction of multiple equivariance variables [67, 17, 18, 24, 54, 34], which we refer to as multivariate equivariance. In the following theorem, we show that multivariate equivariance is provably beneficial since it **monotonically increases the synergy effect** between class information and equivariant prediction, as shown in the following theorem.

**Theorem 3.** *For two transformation variables $A_1, A_2$, we will always have $I(A_1, A_2; C|Z) \geq \max\{I(A_1; C|Z), I(A_2; C|Z)\}$. In other words, multivariate equivariance brings strengthens the explaining-away effect, with a gain of $g = \max\{I(A_2; C|Z, A_1), I(A_1; C|Z, A_2)\}$.*

Theorem 3 can also be easily extended to more equivariant variables. Note that the gains of multivariate equivariance $I(A_2; C|Z, A_1)$ reflects the amount of additional information that the class information $C$ can explain away $A_2$ under the same value of $A_1$; therefore, more diverse augmentations provide a large gain in the synergy effect. Recent works on image world model show that equivariance to multiple transformation delivers better downstream performance and outperforms invariant learning [26].

## 5.3 Model Equivariance

Apart from the design of transformations that is the main focus of E-SSL methods, an often overlooked part is the equivariance of the backbone models, which we call model equivariance. Intriguingly, we find that equivariant networks can be very helpful for E-SSL when *the transformation equivariance aligns well with model equivariance*.

**Setup.** We compare a standard non-equivariant ResNet18 [36] and an equivariant ResNet18 (EqResNet18) *w.r.t.* the $p4$ group (consisting of all compositions of translations and 90-degree rotations) [16] of similar parameter sizes. The models are pretrained on CIFAR-10 and CIFAR-100 for 200 epochs

---

[7]Existing theories rely on strong assumptions between on data augmentation (such as, the augmentation does not change image classes), which is often violated in practice. [58, 68].

with rotation prediction, and then the learned representations are evaluated with a linear probing (LP) head for downstream classification (details in Appendix B). Note that a rotation-equivariant model does not necessarily predict rotation angles perfectly, since in E-SSL, the model only has access to the transformed input but not the ground-truth transformation.

As shown in Table 2 and 4 (see Appendix B), we find that equivariant models bring significant gains for rotation prediction by more than 20% on CIFAR-10 and CIFAR-100. Under aggressive data augmentations (*e.g.,* SimCLR ones), equivariant models provide better equivariant prediction of rotation with high accuracy (91.98% *v.s.* 83.26% on CIFAR-10 and 82.69% *v.s.* 68.29% on CIFAR-100), which also yields better performance on downstream classification with 23.20% and 26.46% higher accuracy respectively. Even more surprisingly, with mild augmentations (no or crop&flip), both models achieve perfect rotation prediction, while equivariant models can still improve classification accuracy a lot.

Therefore, we find that under compatible equivariance, equivariant models have significant advantages for E-SSL in terms of both self-supervised pretraining (better pretraining accuracy) and downstream generalization (best classification accuracy). The following theorem justifies this point by showing that the mutual information *w.r.t.* the transformation $A$ lower bounds the mutual information *w.r.t.* the classification $C$. Therefore, given the same equivariant task (*e.g.,* same data augmentations), features with better equivariant prediction (larger lower bound) will also have more class information.

**Theorem 4.** *For any representation $Z$, its mutual information with the equivariant learning target $A$ lower bounds its mutual information with the downstream task $C$ as follows:*

$$I(Z;A) \leq I(Z;C) - I(X;A|C). \tag{4}$$

Here, a small gap $I(X;A|C)$ means a better generalization between these two tasks. Because $I(X;A|C) = H(A|X,C)$ is a lower bound of $I(A;C|X)$ that indicates class relevance, it further justifies our Principle II (Section 4.2) that better class relevance brings better E-SSL performance.

### 5.4   Strict Equivariant Objectives

Mathematically, an exact definition of equivariance requires that for each transformation $a$ in the input space, there is a corresponding transformation $T_a$ in the representation space so that $f(a(x)) \approx T_a f(x)$. Common rotation prediction objectives do not satisfy this property. Other works also study the use of exact equivariant objectives. Here, we take CARE [34] as an example and compare it against rotation prediction with cross-entropy (CE) loss.

Table 3: Comparison of augmentation-aware and truly equivariant methods (CIFAR-10, ResNet18).

| Equivariant Loss | Train Rot ACC | Test Cls ACC |
|---|---|---|
| CE loss | 97.71 | 57.32 |
| CARE loss | 99.95 | 64.50 |

Table 3 shows that training with exact equivariant objective (CARE) leads to further improvement in test accuracy (57.32% $\rightarrow$ 64.50%), which aligns with our observation of model equivariance in Section 5.3. Altogether they suggest that enforcing exact feature equivariance (either through model architecture or feature regularization) can bring considerable benefits in downstream generalization.

## 6   Conclusion

In this paper, we have provided a general theoretical understanding of how learning from seemingly irrelevant equivariance (such as, random rotations, masks and instance indices) can benefit downstream generalization in self-supervised learning. Leveraging the causal structure of data generation, we have discovered the explaining-away effect in equivariant learning. Based on this finding, we have established theoretical guarantees on how E-SSL extracts class-relevant features from an information-theoretic perspective. We also identify several key factors that influence the explaining-away. Since this work is theory-oriented to fill the gap between practice and theory by investigating how E-SSL works, we do not explore extensively for a better E-SSL design. Nevertheless, the fruitful insights developed in this work could inspire more principled designs of E-SSL methods in future research.

## Acknowledgement

This research was supported in part by NSF AI Institute TILOS (NSF CCF-2112665), NSF award 2134108, and the Alexander von Humboldt Foundation. Yisen Wang was supported by National Key R&D Program of China (2022ZD0160300), National Natural Science Foundation of China (92370129, 62376010), and Beijing Nova Program (20230484344, 20240484642).

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

# A  Omitted Proofs

## A.1  Proof of Proposition 1

*Proof.* It is easy to see that the linear encoder that takes the last $d'$ dimension of the input does not rely on any class information while giving a perfect prediction of $A$, *i.e.,* $f(X) = X_{[d+1:d+d']} = A$. Therefore, it gives random guess prediction on downstream classification. $\square$

## A.2  Proof of Lemma 1

*Proof.* We begin by restating an important result in probabilistic graphical models (PGMs) for conditional independence.

**Lemma 2** (Theorem 3.5 (rephrased) [44]). *For almost all distributions $P$ that factorize over the causal diagram $\mathcal{G}$, that is, for all distributions except for a set of measure zero in the space of CPD (conditional probability distributions) parameterizations, we have that $I(P) = I(\mathcal{G})$, where $\mathcal{I}(\mathcal{G})$ denotes the set of independencies that correspond to $d$-separation:*

$$\mathcal{I}(\mathcal{G}) = \left\{ (\boldsymbol{X} \perp \boldsymbol{Y} \mid \boldsymbol{Z}) : \text{d-sep}_{\mathcal{G}}(\boldsymbol{X}; \boldsymbol{Y} \mid \boldsymbol{Z}) \right\}.$$

Lemma 2 shows that almost all distributions that factorize over the causal diagram $\mathcal{G}$ obey the $d$-separation rules. According to $d$-separation [27, 44], the collider structure in Figure 2 implies that $A \not\perp C|X$ and $A \not\perp C|Z$. Further, recall that for any three random variables $A, B, C$, the mutual information $I(A; B|C) \geq 0$, and $I(A; B|C) = 0$ iff they are conditionally independent, *i.e.,* $A \perp B|C$. Combined the facts above, we will have $I(A; C|X) > 0$ and $I(A; C|Z) > 0$ almost surely. $\square$

## A.3  Proof of Theorem 1

*Proof.* Leveraging Lemma 1, we know that $I(A; C|Z)$ holds almost surely, which implies that

$$H(A|Z) > H(A|Z, C). \tag{5}$$

Since $Z_C$ is a reparameterization of $C$, we have $H(A|Z, C) = H(A|Z, Z_C) = H(A|\tilde{Z})$. Subtracting $H(A)$ on both sides of Eq. (5) gives

$$H(A|Z) - H(A) < H(A|Z_I) - H(A),$$

which is equivalent to $I(A; \tilde{Z}) > I(A; Z)$. Besides, we also have $I(C; \tilde{Z}) = I(C; Z, Z_C) \geq I(C; Z)$, which completes the proof. $\square$

## A.4  Proof of Theorem 2

*Proof.* First, let us consider the first proposition of Theorem 2. We have

$$
\begin{aligned}
I(A; C \mid X) &= I(A; C, X) - I(A; X) \\
&= I(A; C) + I(A; X \mid C) - I(A; X) \\
&= I(A; A + \lambda C \mid C) - I(A; A + \lambda C) \\
&= I(A; A) - I(A; A + \lambda C) \\
&= H(A) - H(A + \lambda C) + H(A + \lambda C \mid A) \\
&= H(A) - H(A + \lambda C) + H(\lambda C).
\end{aligned}
\tag{6}
$$

Since $H(\lambda C) = H(C)$ and $H(A)$ and $H(C)$ are determined by $N_A$ and $N_C$, the goal is then transformed into minimizing $H(A + \lambda C)$. In order to address this problem, we first draw a table below to enumerate the possible outcome of $A + \lambda C$. To determine $H(A + \lambda C)$, we can divide the elements in the table into groups, where they are distributed to the same group if and only if they have the same value. Let us assume that $N_A \leq N_C$ because the opposite condition can be solved in a similar way.

| 0 | 1 | 2 | ... | $N_A - 2$ | $N_A - 1$ |
|---|---|---|---|---|---|
| $\lambda$ | $1 + \lambda$ | $2 + \lambda$ | ... | $N_A - 2 + \lambda$ | $N_A - 1 + \lambda$ |
| $2\lambda$ | $1 + 2\lambda$ | $2 + 2\lambda$ | ... | $N_A - 2 + 2\lambda$ | $N_A - 1 + 2\lambda$ |
| ... | | ... | ... | ... | ... |
| $(N_C - 2)\lambda$ | $1 + (N_C - 2)\lambda$ | $2 + (N_C - 2)\lambda$ | ... | $N_A - 2 + (N_C - 2)\lambda$ | $N_A - 1 + (N_C - 2)\lambda$ |
| $(N_C - 1)\lambda$ | $1 + (N_C - 1)\lambda$ | $2 + (N_C - 1)\lambda$ | ... | $N_A - 2 + (N_C - 1)\lambda$ | $N_A - 1 + (N_C - 1)\lambda$ |

Note that the number of elements in a single group cannot be greater than $N_A$, for at most one element in each column can be distributed to that group. Therefore, we denote $x_k$ as the number of groups that consist of k elements, $k \in \{1, 2, ..., N_A\}$. Our target is $H(A + \lambda C) = \sum_{t=1}^{N_A} x_t \frac{t}{N_A N_C} \ln \frac{N_A N_C}{t}$. For simplicity, we further denote $y_k = \frac{1}{N_A N_C} \ln \frac{N_A N_C}{k}$, then $H(A + \lambda C) = \sum_{t=1}^{N_A} t x_t y_t$.

Considering the total number of elements, we have

$$x_1 + 2x_2 + ... + N_A x_{N_A} = N_A N_C. \tag{7}$$

We now pay attention to the top row and the rightmost column, where the elements are bound to be in different groups. Under closer observation, we find that the element 0 forms a group alone, as does the element $N_A - 1 + (N_C - 1)\lambda$. The element 1 is in a group made up of at most two elements, as are the elements $\lambda, (N_A - 1) + (N_C - 2)\lambda, (N_A - 2) + (N_C - 1)\lambda$. Based on similar analysis, the following relationship can be deduced:

$$x_1 \geq 2, x_1 + 2x_2 \geq 6, ..., x_1 + 2x_2 + ... + (N_A - 1)x_{N_A - 1} \geq N_A(N_A - 1). \tag{8}$$

We also have

$$y_1 > y_2 > ... > y_{N_A}. \tag{9}$$

$$
\begin{aligned}
H(A + \lambda C) &= \sum_{t=1}^{N_A} t x_t y_t \\
&= (N_A N_C - \sum_{t=1}^{N_A - 1} t x_t) y_{N_A} + \sum_{t=1}^{N_A - 1} t x_t y_t \\
&= N_A N_C y_{N_A} + \sum_{t=1}^{N_A - 1} t x_t (y_t - y_{N_A}) \\
&= N_A N_C y_{N_A} + (y_{N_A - 1} - y_{N_A}) \sum_{t=1}^{N_A - 1} t x_t + \\
&\quad (y_{N_A - 2} - y_{N_A - 1}) \sum_{t=1}^{N_A - 2} t x_t + ... + (y_2 - y_3)(x_1 + 2x_2) + (y_1 - y_2)x_1 \\
&\geq N_A N_C y_{N_A} + N_A(N_A - 1)(y_{N_A - 1} - y_{N_A}) + \\
&\quad (N_A - 1)(N_A - 2)(y_{N_A - 2} - y_{N_A - 1}) + ... + 6(y_2 - y_3) + 2(y_1 - y_2) \\
&= N_A(N_C - N_A + 1)y_{N_A} + 2 \sum_{t=1}^{N_A - 1} t y_t.
\end{aligned} \tag{10}
$$

The equality condition for this inequality is

$$x_1 = x_2 = ... = x_{N_A - 1} = 2. \tag{11}$$

This indicates that every secondary diagonal (from upper right to lower left) in the aforementioned table forms a group of elements, which means $\lambda = 1$. This completes the proof of the first claim.

Let us further look at the second proposition. From the discussion above, we know that $\lambda = 1$ is the optimal value, and we want to maximize $I(A, C|X) = H(A) + H(C) - H(A + C)$. Let us assume

that $N_A \geq N_C$. In order to calculate in detail, we first list the probability distribution of $A + C$.

$$P(A + C = 0) = \frac{1}{N_A N_C}, P(A + C = 1) = \frac{2}{N_A N_C}, ..., P(A + C = N_C - 2) = \frac{N_C - 1}{N_A N_C}$$

$$P(A + C = N_C - 1) = \frac{1}{N_A}, P(A + C = N_C) = \frac{1}{N_A}, ..., P(A + C = N_A - 1) = \frac{1}{N_A}, \quad (12)$$

$$P(A + C = N_A) = \frac{N_C - 1}{N_A N_C}, ..., P(A + C = N_C + N_A - 2) = \frac{1}{N_A N_C}.$$

Now, we have

$$
\begin{aligned}
H(A + C) &= -2 \sum_{i=1}^{N_C - 1} \left( \frac{i}{N_A N_C} \ln \frac{i}{N_A N_C} \right) - (N_A - N_C + 1) \frac{1}{N_A} \ln \frac{1}{N_A} \\
&= 2 \sum_{i=1}^{N_C - 1} \left[ \frac{i}{N_A N_C} (\ln N_A + \ln N_C - \ln i) \right] + \frac{N_A - N_C + 1}{N_A} \ln N_A \\
&= \frac{N_C - 1}{N_A} (\ln N_A + \ln N_C) - \frac{2}{N_A N_C} \sum_{i=1}^{N_C - 1} (i \ln i) + \frac{N_A - N_C + 1}{N_A} \ln N_A \\
&= \ln N_A + \frac{N_C - 1}{N_A} \ln N_C - \frac{2}{N_A N_C} \sum_{i=1}^{N_C - 1} (i \ln i).
\end{aligned}
\quad (13)
$$

Thus,

$$
\begin{aligned}
&H(A) + H(C) - H(A + C) \\
&= \frac{N_A - N_C + 1}{N_A} \ln N_C + \frac{2}{N_A N_C} \sum_{i=1}^{N_C - 1} (i \ln i) \\
&\geq \frac{N_A - N_C + 1}{N_A} \ln N_C + \frac{2}{N_A N_C} \int_1^{N_C - 1} x \ln x \, dx \\
&= \frac{N_A - N_C + 1}{N_A} \ln N_C + \frac{1}{N_A N_C} \left[ (N_C - 1)^2 \ln (N_C - 1) - \frac{(N_C - 1)^2 - 1}{2} \right] \\
&= \ln N_C - \frac{1}{N_A} \left[ (N_C - 1) \ln N_C - \frac{(N_C - 1)^2}{N_C} \ln (N_C - 1) + \frac{N_C - 2}{2} \right].
\end{aligned}
\quad (14)
$$

Note that $(N_C - 1) \ln N_C - \frac{(N_C - 1)^2}{N_C} \ln (N_C - 1) + \frac{N_C - 2}{2}$ is greater than 0. Therefore, the lower bound of $I(A, C | X)$ is monotonically non-decreasing with respect to $N_A$. If $N_A$ is adequately large, $I(A, C | X)$ approximates $\ln N_C$. □

### A.5  Proof of Theorem 4

*Proof.* The lower bound can be easily derived by taking the difference between the two quantities:

$$I(Z; C) - I(Z; A) \geq I(Z; C; A) - I(Z; A) \quad (15)$$
$$= - I(Z; A | C) \quad (16)$$
$$\geq - I(X; A | C), \quad (17)$$

where the last line comes from the information processing inequality. □

## B  Experiment Details

In this section, we detail the setting of each individual experiment in this work. All experiments are conducted with a single NVIDIA RTX 3090 GPU.

## B.1 Experiment Details of Different Equivariant Pretraining Tasks

In this experiment, we conduct equivariant pretraining tasks based on seven different types of transformations. In order to maintain fairness and avoid cross-interactions, we only apply random crops to the raw images before we move on to these tasks. We adopt ResNet-18 as the backbone with a two-layer MLP that has a hidden dimension of 2048 and an output dimension corresponding to the pretraining tasks. Under each transformation, we train the model for 200 epochs on CIFAR-10, with batch size 512 and weight decay $10^{-6}$. The detailed pretraining tasks are listed as follows.

**Horizontal Flip & Vertical Flip & Color Inversion & Grayscale.** We randomly (*i.e.,* with probability 0.5) apply the specific transformation to images and require the model to predict whether or not we have really done the transformation. In these cases, the output dimension is 2.

**Four-fold Rotation.** We rotate the images with equal probability (*i.e.,* with probability 0.25) by 0°, 90°, 180°, and 270° and require the model to predict which rotation angle we have actually adopted. In this case, the output dimension is 4.

**Four-fold Blur.** We apply Gaussian blurs to the images using kernel sizes of 0, 5, 9, and 15, where kernel size 0 refers to not applying Gaussian blurs. We then require the model to predict the kernel size. In this case, the output dimension is 4.

**Jigsaw.** We divide the images into $2 \times 2$ patches, randomly shuffle their order, and then require the model to predict the original arrangement. In this case, the output dimension is 24, since there are $4! = 24$ possible permutations for the shuffled arrangements.

During the pretraining tasks, we simultaneously train a classifier, which is a single-layer linear head and is trained without affecting the rest of the network. Apart from the seven tasks, we also conduct a baseline experiment, where we fix a random encoder and optimize the classifier alone in order to assess the effectiveness of these pretraining tasks.

## B.2 Experiment Details of How Class Information Affects Equivariant Pretraining Tasks

In this experiment, our goal is to figure out how class information affects rotation prediction. Figure 4 demonstrates the outline of the model we use to conduct this experiment. We apply random crops with size 32 and horizontal flips with probability 0.5 to the raw images.

**Training objectives.** As for the experiment process, we first use rotation prediction as the pretraining task with a cross-entropy loss between our predicted angles and the actual angles, defined as

$$\mathcal{L}_{rot} = -\frac{1}{N} \sum_{i=1}^{N} \sum_{j=1}^{4} p_{ij} \log(\hat{p}_{ij}), \tag{18}$$

where N is the image number, the one-hot vector $p_{ij}$ refers to the true rotation angle of the $i^{th}$ image, and $\hat{p}_{ij}$ refers to the prediction of the model. In the case where class information is incorporated, we simply add to the original loss function the cross-entropy between the classes predicted by the classifier and their corresponding ground truth labels, defined as

$$\mathcal{L}_{cls} = -\frac{1}{N} \sum_{i=1}^{N} \sum_{j=1}^{C} y_{ij} \log(\hat{y}_{ij}), \tag{19}$$

where N is the image number, C is the class number, the one-hot vector $y_{ij}$ refers to the true class of the $i^{th}$ image, and $\hat{y}_{ij}$ refers to the prediction of the model. In other words, when class information is injected, the loss function is $\mathcal{L}_{rot} + \lambda_1 \mathcal{L}_{cls}$, where the mixing coefficient $\lambda_1$ is a hyper-parameter. Furthermore, to eliminate class information from the first setting, we minimize the classifier loss with $\mathcal{L}_{cls}$, trying to probe class information in the representation; in the meantime, we optimize the encoder to maximize the classification loss, aiming to eliminate any class information that can be found by the classifier. In particular, we adopt a joint training objective for the encoder as $\mathcal{L}_{rot} - \lambda_2 \mathcal{L}_{cls}$, where the mixing coefficient $\lambda_2$ is also a hyper-parameter. This leads to a *min-max optimization* between the encoder and the linear classifier. We choose $\lambda_1 = 0.5$ and $\lambda_2 = 9$, under which the class features can be shown to benefit or harm rotation prediction.

In the pretraining process, we mainly use Resnet-18 as the backbone with a two-layer MLP that has a hidden dimension of 2048 and an output dimension of 4, and a single-layer linear head as

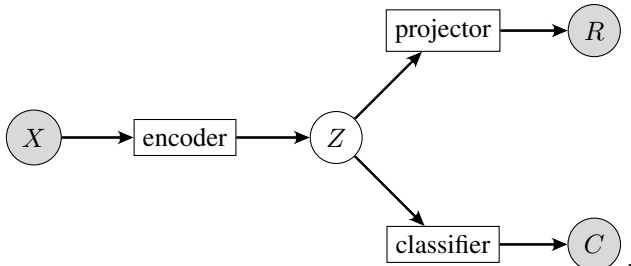

Figure 4: The model of this experiment. $X$: raw input; $Z$: representation; $R$: rotation prediction; $C$: class prediction. For rotation prediction, unless specified, the gradient flowing from the classifier to the encoder is detached.

Table 4: Training rotation prediction accuracy and test linear classification accuracy under different base augmentations (CIFAR-100, ResNet18).

| Dataset | Augmentation | Network | Train Rotation ACC | Test Classification ACC | Gain |
|---------|--------------|---------|--------------------|-------------------------|------|
| CIFAR-100 | None | ResNet18 | 99.83 | 11.31 | |
| | | EqResNet18 | **100.00** | **32.38** | **+21.07** |
| | Crop&Flip | ResNet18 | 90.94 | 13.19 | |
| | | EqResNet18 | **99.88** | **49.47** | **+36.28** |
| | SimCLR | ResNet18 | 68.29 | 10.65 | |
| | | EqResNet18 | **82.69** | **37.11** | **+26.46** |

a classifier. For each setting, we train the model for 200 epochs on CIFAR-10 and CIFAR-100 respectively with batch size 512 and weight decay $10^{-6}$. Additionally, we train the model for 100 epochs on Tiny-ImageNet-200 with batch size 256 and weight decay $10^{-6}$. To further explore different backbones, we also use Resnet-50 and train the model for 200 epochs on CIFAR-10 with batch size 256 and weight decay $10^{-6}$.

Furthermore, we are interested the effects of class information on the accuracy of the pretraining tasks based on other transformations such as horizontal flips and four-fold blurs that are regarded as intrinsically hard. We simply inject class information into the representation by adding a classification loss to the original loss as well. Slightly different from the operation in rotation prediction, we apply random resized crops before conducting the pretraining tasks to avoid interfering with the prediction in these tasks. The other details and parameters are the same as those in rotation prediction.

### B.3 Experiment Details in the study of model equivariance

In order to compare the performance of Resnet and EqResnet, we use rotation prediction as our pretraining task and obtain the linear probing results. We apply various augmentations to the raw images, such as no augmentation, a combination of random crop with size 32 and horizontal flip, and SimCLR augmentation with output size 32. To be more specific, SimCLR augmentation refers to a sequence of transformations, including random resized crop with size 32, horizontal flip with probability 0.5, color jitter with probability 0.8, and finally grayscale with probability 0.2.

In these experiments, we predict rotation angles with a two-layer MLP that has a hidden dimension of 2048 and an output dimension of 4, and a single-layer linear head as a classifier. For each setting, we train the model for 200 epochs on CIFAR-10 and CIFAR-100 with batch size 128 and weight decay $5 \times 10^{-4}$. The results on CIFAR-100 are displayed in Table 4.

## C $\mathcal{V}$-information: Background and Extensions

In this section, we introduce $\mathcal{V}$-information [75], which is a computation-aware and model-aware extension of Shannon's notation that is more suitable for modeling neural representation learning. Then, we extend our theory and show that the main results still hold under $\mathcal{V}$-information.

## C.1 Definitions and Properties of $\mathcal{V}$-information

$\mathcal{V}$-information is proposed by Xu et al. [75] under the consideration of computational constraints, which happens to be one of the drawbacks of traditional Shannon information theory. An additional merit of $\mathcal{V}$-information is that it can be estimated from high-dimensional data. The formal definition of $\mathcal{V}$-information is derived as follows. Denote $Y$ as the target random variable that the model is trying to predict and $X$ as another random variable that provides side information for the prediction of $Y$. Let $\mathcal{X}$ and $\mathcal{Y}$ be the sample spaces of $X$ and $Y$. Define $\Omega := \{f : \mathcal{X} \cup \emptyset \rightarrow \mathcal{P}(\mathcal{Y})\}$ as a set of the functions that maps $X$ to a family of probability distributions over $Y$.

**Definition 1** (Predictive Family). $\mathcal{V} \subseteq \Omega$ is called a predictive family if $\forall f \in \mathcal{V}, \forall P \in range(f)$, $\exists f' \in \mathcal{V}$ that satisfies $\forall x \in \mathcal{X}, f'[x] = P, f'[\emptyset] = P$.

In other words, a predictive family is a set of probability measures that are allowed to be used under computational constraints. The existence of $f'$ indicates that the agent can optionally ignore the side information.

Next, we introduce predictive conditional $\mathcal{V}$-entropy and predictive $\mathcal{V}$-information.

**Definition 2** (Predictive Conditional $\mathcal{V}$-entropy). $H_\mathcal{V}(Y|X) = inf_{f \in \mathcal{V}}\mathbb{E}_{x,y \sim X,Y}[-\log f[x](y)]$. Specifically, $H_\mathcal{V}(Y|\emptyset) = inf_{f \in \mathcal{V}}\mathbb{E}_{y \sim Y}[-\log f[\emptyset](y)]$.

**Definition 3** (Predictive $\mathcal{V}$-information). $I_\mathcal{V}(X \rightarrow Y) = H_\mathcal{V}(Y|\emptyset) - H_\mathcal{V}(Y|X)$.

Apart from the definitions, we have to highlight an important property of predictive $\mathcal{V}$-information.

**Lemma 3** (Xu et al. [75]). $I_\mathcal{V}(A \rightarrow B) = 0$ *iff A and B are independent variables.*

## C.2 Extension to $\mathcal{V}$-information

First, we present the $\mathcal{V}$-information version of Lemma 1.

**Theorem 5** (Explaining-away in E-SSL). *If the data generation process obeys the diagram in Figure 2, then almost surely, A and C is no dependent given X or Z, i.e., $A \not\perp C|X$ and $A \not\perp C|Z$. It implies that $I_\mathcal{V}(C \rightarrow A|X) > 0$ and $I_\mathcal{V}(C \rightarrow A|Z) > 0$ hold almost surely.*

*Proof.* Lemma 3 indicates that for any three random variables $A, B, C$, the inequality $I_\mathcal{V}(A \rightarrow B|C) \geq 0$ always holds and that $I_\mathcal{V}(A \rightarrow B|C) = 0$ iff $A \perp B|C$. Based on the analysis of the collider structure in Appendix A.2, we know that A and C are not independent given either $X$ or $Z$. Thus, we have $I_\mathcal{V}(C \rightarrow A|X) > 0$ and $I_\mathcal{V}(C \rightarrow A|Z) > 0$ almost surely. $\square$

Then, we present the $\mathcal{V}$-information version of Theorem 1.

**Theorem 6.** *Assume that the representation Z consists of two parts $Z = [Z_I, Z_C]$, where $Z_I$ is class-irrelevant, and $Z_C = \phi(C)$ is a representation of the class C with an invertible mapping $\phi$. If there is a positive synergy effect $I_\mathcal{V}(C \rightarrow A|Z_I) > 0$, we will have $I_\mathcal{V}(Z_I \rightarrow A) < I_\mathcal{V}(Z \rightarrow A)$, showing that with class features $Z_C$ we can attain strictly better equivariant prediction. As a consequence, the optimal features of equivairant learning will contain class features.*

*Proof.* We have the assumption $I_\mathcal{V}(C \rightarrow A|Z_I) = H_\mathcal{V}(A|Z_I) - H_\mathcal{V}(A|C, Z_I) > 0$. Given that the function $\phi$ is invertible and $Z_C = \phi(C)$, we have $H_\mathcal{V}(A|C, Z_I) = H_\mathcal{V}(A|Z_C, Z_I) = H_\mathcal{V}(A|Z) < H_\mathcal{V}(A|Z_I)$. Subtracting $H_\mathcal{V}(A)$ from both sides and rewriting the inequality, we finally obtain $I_\mathcal{V}(Z \rightarrow A) > I_\mathcal{V}(Z_I \rightarrow A)$. $\square$

# D Detailed Elaboration on Computing $I(A; C|X)$

The computation of $I(A; C|X)$ requires knowledge of both the pretraining label $A$ and the class label $C$. When both are available, we can compute it following its decomposition $I(A; C|X) = H(A|X) - H(A|X, C)$. Estimating entropy in high-dimensional space is generally hard, and a common approach is to leverage variational estimates for the two entropy terms $H(A|X)$ and $H(A|X, C)$ using neural networks.

**Variational estimates of entropy.** Take $H(A|X)$ as an example. We can learn a NN classifier $P_\theta(A|X)$ optimized by minimizing the cross-entropy $L(\theta) = -\mathbb{E}_{P_d(X,Y)}P_\theta(A|X)$. The following

inequality shows that CE loss is the variational upper bound of $H(A|X)$. When $\theta$ is minimized at the minimum with $P_\theta(A|X) = P_d(A|X)$, we have $L(\theta) = H(A|X)$ as the perfect estimate. Since NNs are generally expressive approximators, we believe that the (converged) CE loss can serve as a good estimate for $H(A|X)$.

**Variational estimates of** $I(A; C|X)$. Combing these two, we notice that a good estimate for $I(A; C|X)$ is the difference between the cross entropy losses of the predictor $P_\theta(A|X)$ and $P_\theta(A|X, C)$, which is exactly what we studied in the verification experiment in Section 4.1. In other words, the gap of the CE losses (or similarly the accuracies) between "rotation" and "rotation+cls", can serve as a quantitative measure of the synergy effect $I(A; C|X)$. A more computationally friendly way is to select only a small subset of samples and to train it shortly. As shown in Figure 3, the gap is already exhibited at the early stage of training.

