# OpenReview forum: "Understanding the Role of Equivariance in Self-supervised Learning"
_NeurIPS.cc/2024/Conference — NeurIPS 2024 poster_

### Official Review · Reviewer_RgKg · 2024-06-22

**Soundness:** 3
**Presentation:** 3
**Contribution:** 2
**Rating:** 5
**Confidence:** 3

**Summary:**

This paper provides a theoretical understanding of equivariant self-supervised learning (E-SSL) methods and their effectiveness. The authors propose an information-theoretic analysis that explains the synergy effect between class information and equivariant transformations, leading to improved downstream performance. They also identify three principles for E-SSL design: lossy transformations, class relevance, and shortcut pruning. The paper further discusses the importance of model equivariance and demonstrates its advantages in E-SSL. Overall, the contributions of this paper lie in the theoretical explanation of E-SSL and the guidance it provides for designing effective E-SSL methods.

**Strengths:**

1. The article is well-structured, and the arguments are clear.
2. The author analyzes how equivariance affects self-supervised learning from an information theory perspective and proposes three principles for designing E-SSL.
3. The theoretical validity is demonstrated through theoretical proofs and some small experiments.

**Weaknesses:**

1. The experiments are conducted on few datasets which is not very convincing.
2. There exist some typos in the paper, such as the “xIn” in line 352.
3. It is better for the authors to provide a design enhancement example and conduct experiments to prove the effectiveness of the example, as it only explains the success of previous E-SSL works.

**Questions:**

1. Is 89.49% mentioned in line 279 the ACC of the SimCLR framework during testing? Even with the enhancement of SimCLR, the task of predicting rotation only achieved a maximum effect of 59.06 in downstream tasks. These two values are not very close.
2. In the paragraph starting from line 328, it is believed that the action space of the instance discrimination task is N, but I think it is actually a combination of many binary classification tasks with an action space of 2. So I think contrastive learning is not a special case to equivariant tasks.
3. Principle 2 states that it is best to be category related, does it mean that category label information needs to be known during pre-training stage in SSL?
4. Why can predictive patches like MAE be considered as an equivariant SSL task?

**Limitations:**

Yes.

---

> ### Author Rebuttal · Authors · 2024-08-07
>
> ##
>
> We thank Reviewer PgKg for appreciating our writing and theoretical contributions. Below we address each of your concerns.
>
> ---
>
> **Q1.** The experiments are conducted on few datasets which is not very convincing.
>
> **A1**. Following your suggestion, we further validate our findings on ResNet-50 and CIFAR-100. The results all agree with our findings in the main paper, that 1) better rotation prediction brings better test error, and 2) model equivariance contributes to better test classification. Notably, model equivariance contributes even more on CIFAR-100.
>
> *Results on CIFAR-100.*
>
> | Augmentation | Network | Train Rotation ACC | Test Classification ACC | Gain |
> | --- | --- | --- | --- | --- |
> | none | ResNet18 | 99.83 | 11.31 |  |
> |  | EqResNet18 | 100 | 32.38 | + 21.07 |
> | crop+flip | ResNet18 | 90.94 | 13.19 |  |
> |  | EqResNet18 | 99.88 | 49.47 | +36.26 |
> | simclr  | ResNet18 | 68.29 | 10.65 |  |
> |  | EqResNet18 | 82.69 | 37.11 | + 26.46 |
>
> *Results on ResNet-50 (crop+flip) on CIFAR-100.*
>
> | Network | Train Rotation ACC | Test Classification ACC |
> | --- | --- | --- |
> | ResNet18 | 90.94 | 13.19 |
> | ResNet50 | 99.30 | 14.53 |
>
> ---
>
> **Q2.** There exist some typos in the paper, such as the “xIn” in line 352.
>
> **A2**. Thanks for pointing out. We will fix them in the revision.
>
> ---
>
> **Q3.** It is better for the authors to provide a design enhancement example and conduct experiments to prove the effectiveness of the example, as it only explains the success of previous E-SSL works.
>
> **A3**. We note that the key focus of this work is to establish the first theoretical understanding of E-SSL, instead of devising a new variant. The former may have a larger significance because it is more fundamental to this field. Driven by our theoretical insights, we also first study **combining model equivariance with training equivariance for E-SSL**, and it improves RotNet by **25.22% on CIFAR-10 and 36.26% accuracy on CIFAR-100**. Remarkably, we are the first to attain comparable performance **by predicting augmentations alone** (while previous methods rely on merging with contrastive learning). This illustrates that our analysis can open doors for new paths for E-SSL designs.
>
> ---
>
> **Q4.** Is 89.49% mentioned in line 279 the ACC of the SimCLR framework during testing?
>
> **A4**. Indeed, we made a typo here. Later, when merging with the model equivariance (Sec 5.3), its performance (82.26%) can be close to SimCLR. We will fix this confusion in the revision.
>
> ---
>
> **Q5**. In the paragraph starting from line 328, it is believed that the action space of the instance discrimination task is N, but I think it is actually a combination of many binary classification tasks with an action space of 2. So I think contrastive learning is not a special case to equivariant tasks.
>
> **A5**. We understand that contrastive learning (CL) can be understood as many binary classification tasks between positives and negatives. Nevertheless, this understanding defines **infinitely many** binary tasks and **every task has been seen only once** during training, a setting rarely studied in the ML literature. The deviation from common learning tasks makes it hard for us to have a solid understanding of how contrastive learning actually behaves.
>
> Instead, it is a common and natural way to understand CL as instance classification (IC). In fact, in the seminal work by Wu et al. [1] (prior to InfoNCE) that **first proposed contrastive learning** (their loss is the same as InfoNCE), they do regard CL as **a non-parametric approach to IC**, as the title suggested. We refer to [1] for a detailed explanation of this connection.
>
> We further note that for a very large number of classes like IC, it is common to subsample the classes in the denominator of CE loss. A well-known example is the **negative sampling technique proposed in word2vec [2]** for learning on a very large vocabulary. In instance classification, existing works also subsample the negative classes [1,3], leading to a mini-batch number of negative samples during training.
>
> Hope this elaboration address your concern! We will clarify this part in the revision.
>
> Ref:
>
> [1] Wu, Zhirong, et al. "Unsupervised feature learning via non-parametric instance discrimination." CVPR. 2018.
>
> [2] Mikolov, Tomas, et al. "Distributed representations of words and phrases and their compositionality." NeurIPS 2013.
>
> [3] Cao, Yue, et al. "Parametric instance classification for unsupervised visual feature learning." NeurIPS 2020.
>
> ---
>
> **Q6**. Principle 2 states that it is best to be category related, does it mean that category label information needs to be known during pre-training stage in SSL?
>
> **A6**. No. For SSL, the labels are not supposed to be known during pretraining. What we are discussing in Principle 2 is that the choice of the augmentation type (eg the rotation or the flip) should be related to the class labels, in the sense that extracting class-relevant information from the input is helpful for predicting these transformations (no class label required) during E-SSL pretraining. We will make it clear in the revision.
>
> ---
>
> **Q7**. Why can predictive patches like MAE be considered as an equivariant SSL task?
>
> **A7**. In this paper, we consider a general notion of equivariant SSL, in the sense that the representations is learned to be aware of the pretext, eg aug parameters. MAE augments by masking patches at random **positions $p$ (the aug parameter in MAE)**. Then, MAE takes the position $p$ as input, and learns to reconstruct images at these locations, thus its representations can adapt to the input position variables $p$, in contrast with contrastive learning whose representation is global and invariant to augmentations. Thus, we regard MAE more as an equivariant method. We will explain it in the revision.
>
> ---
>
> We hope this clarifies your questions. If you would find it satisfactory, we respectfully hope that you may consider re-evaluating our work based on the revisions.

---

> > ### Comment · Reviewer_RgKg · 2024-08-12
> >
> > Thank you for your response. I have checked the rebuttal and tend to keep my score.

---

### Official Review · Reviewer_TYZN · 2024-06-27

**Soundness:** 3
**Presentation:** 2
**Contribution:** 2
**Rating:** 6
**Confidence:** 3

**Summary:**

This paper aims to reduce the gap between theory and practice for equivariant SSL, which refers here to the sub-family of discriminative SSL methods that do not enforce the invariance of representations to augmentations. The authors propose an explanation based on the "explain-away" (_“Explaining away” is a common pattern of reasoning in which the confirmation of one cause of an observed or believed event reduces the need to invoke alternative causes._) and conclude that this principle justifies strictly positive mutual information between the pre-task labels (e.g., the rotation angle) and the downstream task label (i.e., semantic class label) thereby explaining why the pretext and downstream tasks are not systematically misaligned.

**Strengths:**

- Relevance of the research question: while an intuition as to why E-SSL performs better than a random encoder might be easy to formulate (see below in weaknesses), a theoretical formulation supporting these intuitions eludes. A principled explanation would help the community in the design of better and more principles E-SSL approaches.

**Weaknesses:**

My main concern regards the soundness of the contribution, the reasoning leading to this concern follows:
- in this work, the authors use the "explaining-away" principle which relies on the proposed latent variable model shown in Figure 2 to justify E-SSL performance. Authors consider a partitioning of the latent information in $C$ and $S$, the former encapsulates semantic information, the latter the non-semantic information (should hence include the object position & orientation, textures).
- I believe a reasonable intuition as to why E-SSL (e.g., predicting the rotation angle of an image as a pretext task and then performing semantic classification) can perform better than a random encoder - as shown by authors in Figure 1 -, is because of a selection bias in images used for training: each semantic class/object is more likely to be depicted in a specific orientation (e.g., a person standing) than in any other orientation (i.e., a person upside down). This means that the pretext and downstream tasks are not completely misaligned: being aware of the class information allows one to make a better guess about the rotation angle than a random guess. If an object is as likely to be depicted in one orientation than any other orientation than the class label is uninformative to make a guess regarding the rotation angle. The difference in results between horizontal and vertical flip in Figure 1 supports that intuition. The explanation proposed by the authors does not support that intuitive explanation of E-SSL but instead seems to propose a overly-simplified
- The explaining-away principle justify a strictly positive mutual information between $\bar{X}$ and $A$ which authors extend to a strictly positive mutual information between $\bar{C}$ and $A$. It is not obvious from the get-go how authors can theoretically justify this extension.

To summarize, the two aforementioned points should be further discussed by the authors to confirm the soundness of the proposed explanation which seems to take shortcuts both from an intuitive and theoretical perspective. I would suggest discussing these points more thoroughly.

Minor comments:
- Relevance of certain results: Theorem 2 is tied to an over-simplified mixing model ($ X = A + C$)
- Rigor:
     - Notation: notations should be consistent throughout the paper (e.g., line 250 and 256, line 236/237 and line 240) and assumptions should be made clear (e.g., eq (3) and line 293, lambda=1 is not stated)
     - Principle III: refers to theoretical results - "style features may also explain the equivariant target A" - , which have not been explicitly shown (in section 4.1 or in the appendix)
    - Contrastive learning vs. Instance discrimination: while instance discrimination performs classification with a number of classes equivalent to the number of training instances, contrastive learning like SimCLR usually discriminates at a batch level hence I believe line 339 does not accurately describe contrastive learning but rather instance discrimination.
- References for claims: "style features are often easier for NN learning" - line 260,
- Typos: line 352, 68

**Questions:**

- Can you please clarify what is meant by "global" vs. "local" transformations (principle II)?
- Can authors rigorously justify the extension of the explaining-away principle from X to C?

**Limitations:**

Limitations are not explicitly discussed but do mention the use of simplified models (e.g., section 4.3) for theoretical purposes.

---

> ### Author Rebuttal · Authors · 2024-08-07
>
> We sincerely thank Reviewer TYZN for a critical reading of our work. We carefully examine the intuition you propose, and find that it actually fits well into our theory.
>
> ---
>
> **Q1.** Discussion on the intuitive understanding of why E-SSL works and how it fits into our theory.
>
> **A1**. We resonate with your insight that for rotation prediction to be effective, the class/object itself must exhibit a rotational bias (i.e., it is a **necessary** condition). Nevertheless, this intuition lacks a solid theoretical explanation. We find it **naturally fits as a piece of our explaining-away framework, where it can have a rigorous explanation**.
>
> To extend our current analysis, besides the causal diagram in Fig 2, we further consider the inherent rotation angle $\bar A$ and an edge $\bar A\to \bar X$, i.e., the original object is also generated from its intrinsic angle. Notice that now there is a new collider structure $C\to \bar X\leftarrow \bar A$, which means that $I(\bar A;C|\bar X)>0$, which rigorously explains why class information $C$ is helpful for predicting the intrinsic rotation $\bar A$ from the original object $\bar X$.
>
> Since the intrinsic rotation $\bar A$ is unknown, RotNet further applies a (known) random rotation $A$ to $\bar X$ and get $X$, and the rotated image has an angle $A'=\bar A+A$. Apparently, figuring out  $\bar A$ is directly helpful for predicting the random $A$ (the E-SSL target). Formally, we have a collider structure $\bar A\to X\leftarrow A$ and have a positive synergy $I(A;\bar A|X)>0$.
>
> Thus, our explaining-away analysis justifies that 1) knowing $C$ is also helpful for predicting $\bar A$, and 2) knowing $\bar A$ is helpful for predicting $A$ as well. In this way, introducing $\bar A$ does not invalidate our previous theory but provide **more fine-grained analysis on the way of how class information helps rotation prediction**.
>
> Thank you again for this valuable perspective. We will definitely incorporate it and acknowledge your insights.
>
> ---
>
> **Q2**. Theorem 2 is tied to an over-simplified mixing model ($X=A+\lambda C$).
>
> **A2**. We note that Theorem 1 already gives us **a general characterization that is agnostic of the data distribution or the model class**. Therefore, our framework and analysis does apply to general real-world scenarios. Due to its generality, it is also hard to obtain a very fine-grained quantitative analysis. Thus, the purpose of Thm 2 is to gain more quantitative insights with a linear model. Note that linear data assumptions are also commonly adopted in SSL theory, eg [1]. We leave more complex cases if left for future work.
>
> **Ref:**
>
> [1] Wen, Zixin, and Yuanzhi Li. "Toward understanding the feature learning process of self-supervised contrastive learning." ICML. PMLR, 2021.
>
> ---
>
> **Q3**. Rigor:
>
> > Notation: notations should be consistent throughout the paper (e.g., line 250 and 256, line 236/237 and line 240) and assumptions should be made clear (e.g., eq (3) and line 293, lambda=1 is not stated)
>
>
> Thanks. We will fix it.
>
> > Principle III: refers to theoretical results - "style features may also explain the equivariant target A" - , which have not been explicitly shown (in section 4.1 or in the appendix)
>
>
> Indeed, we didn’t elaborate the style features here. Similar to the intrinsic rotation variable $\bar A$, the feature variable $S$ also constitutes the object $\bar X$ (with a causal edge $S\to \bar X$). Due to the explaining-away effect in the collider structure $S\to X\leftarrow A$, $S$ can also explain and help predict $A$ with $I(S;A|X)>0$. Therefore, style features, as an easy-to-learn shortcut, can lead the model to learn fewer class features. We will elaborate this relationship rigorously in the revision.
>
> > Contrastive learning vs. Instance discrimination
>
>
> We note that contrastive learning can be seen as a minibatch approximation to instance classification by random subsampling a few instances (e.g., 2048) at each time. In expectation, the objective is still distinguishing the positive instance from all the other negative instances in the dataset. **This negative class sampling technique is widely used in standard classification tasks, e.g., negative sampling in word2vec** [1]. Thus, we think it is plausible to regard contrastive learning essentially as an instance classification task.
>
> References:
>
> [1] Mikolov et al. Distributed Representations of Words and Phrases and their Compositionality. NeurIPS 2013.
>
> ---
>
> **Q4**. Reference for claims and Typos.
>
> **A4**. Thanks. We will add the references and fix the typos.
>
> ---
>
> **Q5**. Can you please clarify what is meant by "global" vs. "local" transformations (principle II)?
>
> **A5**. Here, local changes refer to those that only modify pixel values but do not modify the pixel’s positions (eg color inversion). Therefore, to predict these operations, the NNs only need to look at each pixel’s values without relying on global context. Instead, global operations change pixel positions (e.g., rotation, flip), and in these cases, figuring out the global content can facilitate the prediction of these global changes. We will clarify it in the revision.
>
> ---
>
> **Q6**. Can authors rigorously justify the extension of the explaining-away principle from X to C?
>
> **A6**. Of course. In fact, according to probabilistic graphical models, **as long as there is a collider structure between the variables**, there will be an explaining-away effect. In the causal structure $C\to \bar X\to X$, we can also omit the intermediate node $\bar X$ (which also holds since $X$ is causally dependent on $C$) and write the collider structure $C\to X\leftarrow A$, which implies the explaining-away effect with $I(A;C|X)>0$.
>
> ---
>
> Thank you for your careful reading and for sharing your insights. If you would find our explanations satisfactory, we respectfully ask you to consider re-evaluating our work based on the revisions. We are also happy to address any remaining concerns during the discussion stage.

---

> > ### Comment · Reviewer_TYZN · 2024-08-10
> > **Answer to Rebuttal**
> >
> > Thank you to the authors for their efforts in addressing my comments.
> > I have re-read the paper and read the other reviews/rebuttals.
> > I am happy to increase for the following reasons:
> > - authors have cleared my concerns regarding the soundness of their theoretical results (i.e., use of explaining away effect)
> > - going through the material once again, I see better how my intuition can be explained by the author's work, I believe connecting the author's contributions with a higher level intuition the reader might have regarding E-SSL and why it works would benefit the paper and hope the authors will adjust the manuscript accordingly.
> >
> > Overall, I believe the paper contributes to a better understanding of E-SSL methods.
> > I believe the manuscript should be updated to incorporate the distinction between CL and ID made by the authors in the rebuttal for completeness.

---

### Official Review · Reviewer_yP9X · 2024-06-28

**Soundness:** 3
**Presentation:** 3
**Contribution:** 3
**Rating:** 7
**Confidence:** 4

**Summary:**

This study contributes theoretical insights that enhance our understanding of conventional practices in SSL training. Building on these theoretical foundations and supported by experimental evidence, the study puts forward several principles for the practical implementation of equivariant self-supervised learning designs.

**Strengths:**

This study presents theoretical findings corroborated by experimental results. Specifically, the theory enhances our understanding of the design of augmentation functions. These findings are crucial as previous designs of augmentation functions have largely relied on empirical exploration.

**Weaknesses:**

Experiments are conducted using smaller-scale models.

Typo in line 352.

**Questions:**

Could the principles 1,2,3 be applied to other domains beyond vision-based SSL?

**Limitations:**

Please refer to the question.

---

> ### Author Rebuttal · Authors · 2024-08-07
>
> We thank Reviewer yP9X for acknowledging our contributions to theoretical understandings. Below, we further address your concerns on the empirical side.
>
> ---
>
> **Q1.** Experiments are conducted using smaller-scale models.
>
> **A1.** Thank for your advice! The experiments are mainly designed to validate our theoretical insights, and as shown in previous works, E-SSL methods generalize well across different model sizes. Following your suggestions, we further validate our findings on a larger model ResNet-50 and a more complex dataset CIFAR-100.
>
> As shown below, the results agree well with our findings in the main paper, that 1) better rotation accuracy under a larger model brings better test classification accuracy; and 2) model equivariance brings significant gains. Notably, the improvement of model equivariance is even more significant on CIFAR-100, considering that CIFAR-100 has 100 classes whose accuracy is harder to improve.
>
> *Results on CIFAR-100 (akin to Table 2 on CIFAR-10).*
>
> | Augmentation | Network | Train Rotation ACC | Test Classification ACC | Gain |
> | --- | --- | --- | --- | --- |
> | none | ResNet18 | 99.83 | 11.31 |  |
> |  | EqResNet18 | 100 | 32.38 | + 21.07 |
> | crop+flip | ResNet18 | 90.94 | 13.19 |  |
> |  | EqResNet18 | 99.88 | 49.47 | +36.26 |
> | simclr  | ResNet18 | 68.29 | 10.65 |  |
> |  | EqResNet18 | 82.69 | 37.11 | + 26.46 |
>
> *Results on ResNet-50 (with crop+flip) on CIFAR-100.*
>
> | Network | Train Rotation ACC | Test Classification ACC |
> | --- | --- | --- |
> | ResNet18 | 90.94 | 13.19 |
> | ResNet50 | 99.30 | 14.53 |
>
> ---
>
> **Q2**. Could the principles 1,2,3 be applied to other domains beyond vision-based SSL?
>
> **A2**. Yes! Our principles are based on a theoretical analysis of the **general methodology of E-SSL**, not tailored to vision-based SSL. In other domains (text, graph, speech, time series etc), there are also many similar SSL methods that can be understood as E-SSL in the general sense (see e.g., the discussion of MAE and BERT in Sec 5.1). Taking the **BERT model in NLP** as example, our theory can understand its success from these the three principles:
>
> - **Principle I:** First, BERT learns representations by predicting the masked words (task $A$), and it is shown that a large masking ratio is important for learning useful features (for task $C$). As analysed in Principle I, this is also because we need to corrupt the input enough in order for the high-semantic class-related features to be utilized for masked prediction.
> - **Principle II:** Meanwhile, class information (high-level text semantics) is indeed helpful for masked prediction, thus demonstrating the importance of class relevance (Principle II).
> - **Principle III:** At last, masking prediction as a token-level dense prediction task helps prevent the shortcut features (Principle III).
>
> Thus, the insights of our principles are general and not limited to a certain modality. As the first theoretical work in the field of E-SSL, it can help explain many existing E-SSL-like approaches as well as inspire new designs in multiple domains.
>
> ---
>
> Hope the explanation above address your concerns! Please let us know if there is more to clarify.

---

> > ### Comment · Reviewer_yP9X · 2024-08-09
> >
> > Thank you for conducting the additional experiments and providing the detailed discussions in response to the question.
> > The authors' rebuttal has addressed and resolved previously raised concerns.

---

> > > ### Author Response · Authors · 2024-08-09
> > > **Thanks**
> > >
> > > Thank you for the prompt response! We are very glad to hear that your concerns are now resolved in our rebuttal. We will be sure to incorporate these discussions in our revision.

---

### Official Review · Reviewer_a2cS · 2024-07-12

**Soundness:** 3
**Presentation:** 4
**Contribution:** 3
**Rating:** 7
**Confidence:** 4

**Summary:**

This paper proposes a theoretical and empirical study of the role of invariant and equivariant representations in self-supervised learning. While a number of works have been focused on learning equivariant representations, it remains unclear whether or not equivariance is beneficial in specific tasks. By studying how applying/predicting transformations in latent space is a task that requires class information (through the explaining away effect) the authors derive insights on the behaviour of existing methods and provide directions for future work, notably that invariance may not be necessary at all to obtain good representations.

**Strengths:**

The theoretical analysis confirms previous empirical findings, for example that the considered augmentations must be complex/lossy [1] as well as diverse [1] to improve how much class information is present in the representations.

The collider structure and explaining away phenomenon are an elegant way to understand the interplay between the original data and its transformations, showing that C and A cannot be considered to be independent.

While invariance loss are commonly used to learn equivariant representations[2,3] (in conjunction with an equivariant objective), previous work had found that even with a purely predictive objective, competitive performance can be obtained[1,4]. This is in line with the conjecture line 281 "learning from equivariance alone can achieve competitive performance", and raises an important future line of work.


[1]Garrido, Quentin, et al. "Learning and leveraging world models in visual representation learning." arXiv preprint arXiv:2403.00504 (2024).

[2] Gupta, Sharut, et al. "Structuring representation geometry with rotationally equivariant contrastive learning." arXiv preprint arXiv:2306.13924 (2023).

[3]Devillers, Alexandre, and Mathieu Lefort. "Equimod: An equivariance module to improve self-supervised learning." arXiv preprint arXiv:2211.01244 (2022).

[4] Garrido, Quentin, Laurent Najman, and Yann Lecun. "Self-supervised learning of split invariant equivariant representations." arXiv preprint arXiv:2302.10283 (2023).

**Weaknesses:**

The paper seems to focus on two distinct families of methods. Methods which are augmentation aware (e.g. RotNet) and methods which are truly equivariant, which are explicitly designed to learn the transformation between transformed data in latent space, bringing more structured representations. While this distinction may not affect the general ideas and theoretical arguments, it may impact empirical results and practical considerations. It would be good to have a discussion/comparison of these distinct approaches.

Most of the practical insights stemming from this work are qualitative. A conclusion such as "knowing C is beneficial to predicting A" is a useful general guideline, but doesn't directly translate to practical guidance. The results would be stronger with more quantitative assessments of class relevance (perhaps in another, more computationally friendly domain), i.e. estimating I(A;C|X). This should be seen more as an avenue for future work though.

**Questions:**

In experiments such as figure 1, what is the training loss/method considered ?

Theorem 1 puts forward an argument which seems key to the success or not of equivariant SSL (when used without an invariant objective), "How much information about C do you need to know to predict A". Bringing this point (and class relevance in general) forward in section 3.1 would make the motivation from figure 1 clearer.

**Limitations:**

Limitations are addressed in section 6, focusing on the theoretical nature of the work and how it doesn't directly translate into improved methods. It would be good to elaborate further on the limitation of the analysis, such as how different instantiations of an equivariant object may lead to different behaviors, or the difficulty of characterizing quantities such as I(A;C|X) for various transformations.

---

> ### Author Rebuttal · Authors · 2024-08-07
>
> We thank Reviewer a2cS for acknowledging our theoretical contributions. Below we address your main concerns.
>
> ---
>
> **Q1**. Comparing augmentation aware methods (e.g. RotNet) and truly equivariant methods (eg CARE).
>
> > While this distinction may not affect the general ideas and theoretical arguments, it may impact **empirical results and practical considerations**
> >
>
> **A1**. Thank you for your insightful question. For a controlled study on the role of true equivariance, we add the equivariant objective proposed in CARE [1], which is known to enforce true equivariance when it attains an optimum. The results are shown below.
>
> We find that although the two have similar rotation loss, training with the equivariance regularization leads to further improvement in test accuracy (57.32% → 64.50%), which is quite similar to our findings on the model equivariance.  As discussed in Sec 5.3, these pieces of evidence suggest that **enforcing feature equivariance (either through model architecture or feature regularization) does have considerable benefits compared to vanilla predictive loss**. This is a valuable insight and we will add it in the revision. Thanks for bringing it up!
>
> |  | Rotation ACC | Equi loss (Equivariance) | Test Classification ACC |
> | --- | --- | --- | --- |
> | RotNet | 97.71 | 18.8444 | 57.32 |
> | RotNet + 0.1*Equi Loss | 99.95 | 0 | 64.50 |
>
> ---
>
> **Q2**. Quantitative estimate of I(A;C|X).
>
> **A2**. Indeed, in our analysis, the mutual information I(A;C|X) is the key factor that upper bounds the utility of class features during pretraining and thus determines the downstream transferability of E-SSL. In fact, the performance gap between “rotation” and “rotation+cls” shown in the controlled experiments in Sec 4.1 can serve as a surrogate metric for the MI as a variational estimate.
>
> Due to the limit of space, we put the detailed derivation in a following comment that you may read if interested. Please let us know if there is more to clarify.
>
> ---
>
> **Q3.** In experiments such as figure 1, what is the training loss/method considered?
>
> **A3**. For a fair comparison, we follow the standard RotNet training setting and adopt CE loss for predicting discrete variables. We adopt the same training hyperparameters for all settings. More details are described in Appendix B.1. We will make it more clear in revision.
>
> ---
>
> **Q4**. Theorem 1 puts forward an argument which seems key to the success or not of equivariant SSL (when used without an invariant objective), "How much information about C do you need to know to predict A". Bringing this point (and class relevance in general) forward in section 3.1 would make the motivation from figure 1 clearer.
>
> **A4**. Thank you for your suggestion. During our writing, we felt that although important, it was hard to bring out the “how C helps to predict A” perspective earlier in Sec 3.1 without explaining the explaining-away effect (that appears later in Sec 4.1). Though it may sound natural after understanding the explaining-away effect, before that, people might wonder, “why do you care about how C helps to predict A” (since this retrospective understanding is also new). Bearing this in mind, we did not want to introduce puzzles here. We will try to re-organize it for a better balance. Thanks again for sharing your thoughts!
>
> ---
>
> Thank you again for your insightful questions, which helps make this work more complete. Please do not hesitate to let us know if you have additional concerns.

---

> > ### Comment · Reviewer_a2cS · 2024-08-08
> >
> > Thank you for the clarifications and experiments with CARE.
> > The experiments with CARE as well as the approximation of $I(A;C | X)$ are welcome as they help relate the theoretical insights to practical E-SSL methods as well as help in designing them.
> >
> > I hope that all discussions can be added in a revised version of the manuscript.

---

> > > ### Author Response · Authors · 2024-08-09
> > > **Thanks**
> > >
> > > Thank you for the prompt response and for appreciating our new experiments and discussions. We concur with you that these results would enhance the theoretical and empirical insights of the proposed understanding. We will be sure to incorporate these results in our revision.

---

> ### Author Response · Authors · 2024-08-07
> **Detailed elaboration on computing $I(A;C|X)$**
>
> The computation of I(A;C|X) requires knowledge of both the pretraining label $A$ and the class label $C$. When both are available, we can compute it following its decomposition $I(A;C|X)=H(A|X)-H(A|X,C)$. Estimating entropy in high-dimensional space is generally hard, and a common approach is to leverage variational estimates for the two entropy terms $H(A|X)$ and $H(A|X,C)$ using neural networks.
>
> **Variational estimates of entropy**. Take $H(A|X)$ as an example. We can learn a  NN classifier $P_\theta(A|X)$ optimized by minimizing the cross-entropy $L(\theta)=-E_{P_d(X,Y)} P_{\theta}(A|X)$. The following inequality shows that CE loss is the variational upper bound of $H(A|X)$. When $\theta$ is minimized at the minimum with $P_\theta(A|X)=P_d(A|X)$, we have $L(\theta)=H(A|X)$ as the perfect estimate. Since NNs are generally expressive approximators, we believe that the (converged) CE loss can serve as a good estimate for $H(A|X)$.
>
> $L(\theta)=H(A|X)+KL(P_d(A|X)\|P_\theta(A|X))\geq H(A|X)$
>
> **Variational estimates of** $I(A;C|X)$. Combing these two, we notice that a good estimate for $I(A;C|X)$ is the difference between the cross entropy losses of the predictor $P_\theta(A|X)$ and $P_\theta(A|X,C)$, which is exactly what we studied in the verification experiment in Sec 4.1. In other words, the gap between the CE losses (or similarly the accuracies) between “rotation” and “rotation+cls”, can serve as a quantitative measure of the synergy effect $I(A;C|X)$. A more computationally friendly way is to select only a small subset of samples and to train it shortly. As shown in Figure 3, the gap is already exhibited at the early stage of training.

---

### Author Rebuttal · Authors · 2024-08-07

We thank all reviewers for their positive comments and constructive critics on our work. Taking these valuable insights into consideration, we address these problems carefully in each response. We will also do the following:

- **A series of validation experiments on more datasets and models.** We will add a series of new experiments that reproduce our verification experiments on more datasets (CIFAR-100 and Tiny-ImageNet-200) and larger networks (ResNet-50), including *the pretext comparison (Fig 1),  the controlled study on class features (Fig 3), the base augmentation comparison (Tab 1) and model equivariance (Tab 2)*. See Figs A & B and Tabs A & B in the **rebuttal PDF** for details.
- **Fix typos.** We will improve readability by fixing the typos pointed out by the reviewers.
- **Discuss the Role of rotational bias.** Inspired by Reviewer TYZN on the role of intrinsic rotational bias, we will further incorporate the intrinsic equivariance variable $\bar A$ into the causal graph and elaborate on the explaining-away effect of $\bar A$ (elaborated below) for the prediction of class $C$. Notably, **this is a natural extension of our theory and does not change the overall framework, understanding, or message of E-SSL, but makes this understanding more complete**. We sincerely acknowledge Reviewer TYZN for sharing the insight.
- **Compare equivariant to augmentation-aware methods**. Inspired by Reviewer a2cS, we will add the comparison between truly equivariant and augmentation-aware methods, showing that true feature equivariance has additional benefits.

##

---

### Decision · Program_Chairs · 2024-09-25

**Decision:**

Accept (poster)

**Comment:**

This paper explores why and how equivariant self-supervised learning (e.g., rotation prediction) learns useful representations from an information-theoretic perspective. The authors effectively addressed the reviewers' concerns during the author-reviewer discussion phase, and after the discussion, all the reviewers unanimously support the acceptance of this paper. Although AC has some concern about the limited empirical support, AC thinks the merits of this paper outweigh the concern: this paper can be a useful guideline for researchers considering equivariant self-supervised learning techniques. Therefore, AC recommends acceptance.